# Improved maps of surface water bodies, large dams, reservoirs, and lakes in China

Xinxin Wang[1, 2], Xiangming Xiao[2, *], Yuanwei Qin[2], Jinwei Dong[3], Jihua Wu[4], Bo Li[1, 5, *]

[1] Ministry of Education Key Laboratory for Biodiversity Science and Ecological Engineering, National Observations and Research Station for Wetland Ecosystems of the Yangtze Estuary, Institute of Biodiversity Science and Institute of Eco-Chongming, School of Life Sciences, Fudan University, Shanghai 200438, China. [2] Department of Microbiology and Plant Biology, Center for Earth Observation and Modeling, University of Oklahoma, Norman, OK 73019, USA. [3] Key Laboratory of Land Surface Pattern and Simulation, Institute of Geographic Sciences and Natural Resources Research, Chinese Academy of Sciences, Beijing 100101, China. [4] State Key Laboratory of Grassland Agro-Ecosystems, and College of Ecology, Lanzhou University, Lanzhou, Gansu 730000, China. [5] Centre for Invasion Biology, Institute of Biodiversity, Yunnan University, Kunming, Yunnan 650504, China

Corresponding authors: Xiangming Xiao, xiangming.xiao@ou.edu; Bo Li, bool@fudan.edu.cn

## Abstract

Data and knowledge of surface water bodies (SWB), including large lakes and reservoirs (surface water areas > 1 km$^2$) are critical for the management and sustainability of water resources. However, the existing global or national dam datasets have large georeferenced coordinate offsets for many reservoirs, and some datasets have not reported reservoirs and lakes separately. In this study, we generated China's surface water bodies, Large Dams, Reservoirs, and Lakes (China-LDRL) dataset by analyzing all available Landsat imagery in 2019 (19,338 images) in Google Earth Engine and very-high spatial resolution imagery in Google Earth Pro. There were ~3.52 × 10$^6$ yearlong SWB polygons in China for 2019, only 0.01 × 10$^6$ of them (0.43%) were of large size (> 1 km$^2$). The areas of these large SWB polygons accounted for 83.54% of the total 214.92 × 10$^3$
km$^2$ yearlong surface water area (SWA) in China. We identified 2,140 large dams, including 1,494
reservoir dams and 646 river dams, 1,976 large reservoirs (16.42 × 10$^3$ km$^2$), and 3,508 large lakes
(75.97 × 10$^3$ km$^2$). In general, most of the dams and reservoirs in China were distributed in South
China, East China, and Northeast China, whereas most of lakes were located in West China, the
Lower Yangtze River Basin, and Northeast China. The provision of the reliable, accurate China-
LDRL dataset on dams, large reservoirs and lakes will enhance our understanding of water
resources management and water security in China. The China-LDRL dataset is publicly available
at https://doi.org/10.6084/m9.figshare.16964656.v2 (Wang et al., 2022).
**1. Introduction**

Surface water bodies (SWB), including large lakes and reservoirs (surface water areas > 1

km$^2$), play an important role in the control and management of water resources (Yang and Lu, 2014,
2013; Feng et al., 2013, 2019). A reservoir is usually defined as artificial lake formed by
constructing dams across rivers (impoundment reservoir) (Thornton et al., 1996; Hayes et al., 2017)
or partially or completely formed by enclosed waterproof banks with concrete or clay (off-stream
reservoir) (Xiang et al., 2019; Thornton et al., 1996). Off-stream reservoirs usually include
mountain and plain reservoirs (**Fig. 1**). Nearly 50% of the global large dams were built primarily
for agricultural irrigation through storing, regulating, and diverting water (Mulligan et al., 2020).
Additionally, they are also used for hydropower generation, human and industrial uses, and flood
peak attenuation (Lehner et al., 2011; Lehner and Döll, 2004; Wang et al., 2021a). Large lakes
have been the subject of great interest not only because of their water resources but also as

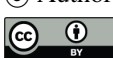



indicators of local climate change and anthropogenic activities (Birkett and Mason, 1995; Ma et
al., 2011), and they could provide vital ecosystem services for human being, such as alteration of
river flow, supplies of irrigation water, fisheries, and abundant valuable mineral deposits, and have
disproportionate effects on the global carbon cycle (Ran et al., 2021; Armstrong, 2010; Ma et al.,
2011). Improved understanding of the detailed distributions of SWB, large dams, reservoirs, and
lakes could provide crucial information on water resources, environmental health, status of
ecosystems, and agricultural sustainability (Lehner and Döll, 2004).

**Insert Fig. 1 here**

China has the largest population, fastest-growing economy, increased expansion of irrigation,

relatively scarce water, dated infrastructure, and inadequate governance (Liu and Yang, 2012;
Wang et al., 2020a; Tao et al., 2020). China encompasses almost 20% of the world's population
but contains only 7% of the world's fresh water, and as the result, it has much smaller fresh water
resource per capital than do most other countries (Feng et al., 2019; Dalin et al., 2014). Since 1980s,
China has taken diverse measures to ensure the long-term water security (Zhou et al., 2020). For
example, China has a remarkable increase of reservoir construction across the country (Wang et
al., 2021a), and the total dam number increased to ~89,700 by 2008 in China (Yang and Lu, 2014).
The Three Gorges Reservoir, which is the world's largest hydroelectric dam (Three Gorges Dam),
is fully operational for flood control, power generation, navigation, and water use (Wu et al., 2004;
Zhang et al., 2012; Wang et al., 2013, 2020a). China also has a large number of lakes with
tremendously cultural and economic importance. Previous study reported that there were 2,693





large lakes (area > 1 km$^2$) in China during 2005-2006, covering 0.9% of China's land area (Ma et
al., 2011). However, due to intensive human activities and climate change over the last three
decades, several natural lakes have converted into reservoirs, accelerating dramatically shrinkage
of lake areas (Yang and Lu, 2014; Ma et al., 2011). Therefore, the improved information on the
distribution of large reservoirs and lakes in China is needed for assessing the impact of human
activities and climate change on SWB, water management, and water security in China (Yang and
Lu, 2014).

There are several published global dam and reservoir datasets that include information from

China (**Table 1**). The World Register of Dams (WRD), which was published by the International
Commission on Large Dams (ICOLD, 2011), is the largest and widely-used dataset (Mulligan et
al., 2020; Paredes-Beltran et al., 2021; Wang et al., 2021a). It reports 23,841 dam entries for China,
however, a large proportion of those entries are not georeferenced with latitude and longitude
information, which limits its wide application (Wang et al., 2021a). The GlObal geOreferenced
Database of Dams (GOODD) V1 dataset reported 9,234 georeferenced dams in China (Mulligan
et al., 2020), however, the information (e.g. area, volume capacity) of all the corresponding
reservoirs was not reported. The FAO's (Food and Agriculture Organization of the United Nations)
global information system on water resources and agricultural water management (AQUASTAT)
lists 14,000 dams in the world, but only part of 722 dams in China were georeferenced, and has
not been updated since 2015. The Global Reservoir and Dam database (GRanD), developed by the
Global Water System Project (GWSP), compiled the available reservoir and dam information
globally (Lehner et al., 2011) and has been updated for the year 2019. However, it only lists 922
geolocated dam entries for China. Recently, Wang et al. (2021a) released a global Georeferenced
global dam and reservoir (GeoDAR) dataset with 5,347 georeferenced dams in China, and the
reservoirs had more than 40 attributes acquired from the WRD dataset. However, our preliminary
quality-check of the dataset shows that the georeference information of many dams in the GeoDAR
dataset has moderate to substantial shifts (or offsets, mis-location), up to 500m or more (**Fig. S1**),
indicating further improvement is needed before it can be used for geospatial analysis. There were
also some published dam and reservoir maps at the national scale (**Table. 1**), but these maps neither
included georeferenced dams nor reported reservoir attributes (e.g. reservoir area).
**Table 1. Information on published dam and reservoir datasets for the globe and China.**

| Name | Spatial domain | Number of dams in the globe | Number of dams in China | Georeferenced dam? | Reservoir information (area …)? |
|---|---|---|---|---|---|
| WRD | Global | ~ 60000 | 23,841 | Either not georeferenced or inaccessible. | Yes, > 40 attributes |
| GOODD V1 | Global | 38667 | 9,231 | Yes | No |
| FAO AQUASTAT | Global | 14000 | 722 | Partly georeferenced | Yes, reservoir capacity and area |
| GRanD | Global | 7320 | 922 | Yes | Yes, ~ 50 attributes |
| GeoDAR | Global | 23680 | 5,345 | Yes | Yes, attributes from WRD dataset |
| CLRM | China | / | 89,700 | No | Yes, reservoir capacity and area |
| BFNCW | China | / | 98,002 | No | No |

WRD: the World Register of Dams (https://www.icold- cigb.org); GOODD: GlObal geOreferenced Database of
Dams (Mulligan et al., 2020); FAO AQUASTAT: The Food and Agriculture Organization of the United Nations



(FAO) global information system on water resources and agricultural water management
(http://www.fao.org/aquastat/en/databases/dams/); GRanD: the Global Reservoir and Dam database (Lehner et
al., 2011); GeoDAR: Georeferenced global dam and reservoir dataset (Wang et al., 2021a); CLRM: China's
Lakes and Reservoirs Map (Yang and Lu, 2014); BFNCW: Bulletin of First National Census for Water from
Ministry of Water Resources the People's Republic of China (http://www.mwr.gov.cn/2013pcgb/index.html). "/"
means these data were published in China, but global dam information is unavailable.

In addition to the dam and reservoir datasets, several studies have reported the spatial

distribution and multi-year dynamics of inland SWB (Tao et al., 2020; Ma et al., 2011; Wang et
al., 2020a; Feng et al., 2019) and lakes (Gao, 2015; Gao et al., 2012; Yang and Lu, 2014; Ma et al.,
2011) in China, however, they did not explicitly explore the spatial distribution of large reservoirs
and lakes in China, making it impossible to assess the impact of human activities on these two
types of water resources (Yang and Lu, 2014). Thus, to date, the spatial distributions of SWB, large
dams, reservoirs, and lakes in China have not been fully investigated and documented, yet.

The objective of this study was to produce detailed and accurate maps of open SWB, large

dams, reservoirs, and lakes (surface water area > 1 km$^2$) in China in 2019, the latest year when this
study started in late 2020. First, this study used time-series Landsat imagery in 2019 and Google
Earth Engine (GEE) cloud computing platform as well as the simple and robust surface water
mapping algorithm (Zou et al., 2018, 2017; Zhou et al., 2019b; Wang et al., 2020a) to generate
raster maps of SWB in China at 30-m spatial resolution. Second, we converted the raster map of



SWB to a vector map of SWB and identified those large SWB with area > 1 km$^2$. Third, we
combined the vector map of SWB with the historical satellite images in 2019 within China in
Google Earth Pro to identify dams and released China's surface water bodies, large dams,
reservoirs, and lakes dataset, namely, China-LDRL. Forth, we analyzed the spatial distribution of
SWB, large dams, reservoirs, and lakes in China. Finally, we discussed the reliabilities,
uncertainties, limitations, outlooks, and implications of the China-LDRL dataset.

**2. Materials and Methods**
**2.1 Study area**
The study area covered all the provincial-level administrative divisions in China (**Fig. 2a**),
including 23 provinces, 2 special administrative regions (Hong Kong and Macao), 4 municipalities
(Beijing, Tianjin, Shanghai, and Chongqing), and 5 Autonomous Regions (Inner Mongolia,
Guangxi, Tibet, Ningxia, and Xinjiang). Since Macao and Hong Kong have relatively small areas
and are very close to Guangdong Province, we combined them as one region (Guangdong) when
we performed the statistical analysis in this study.
**2.2 Data**
**2.2.1 Landsat data**
In this study, we used the available Landsat surface reflectance (SR) images in the GEE



platform, and there was a total of 19,338 images in 2019 for China, including 9,028 Landsat-7
ETM+ images and 10,310 Landsat-8 OLI images (~21.73 TB). The detailed information of
Landsat SR products is available on the GEE platform (https://developers.google.com/earth-
engine/datasets/catalog/landsat, last access: 18 February 2022). All these images had undergone
necessary pre-processing in GEE, including radiometric calibration and atmospheric correction.
We used the quality assurance (QA) band that was generated by the CFMASK algorithm (Zhu et
al., 2015) to identify bad-quality observations, including clouds and cloud shadows (Murray et al.,
2019; Pekel et al., 2016). We also used the Shuttle Radar Topography Mission (SRTM) digital
elevation model (DEM) data, the solar azimuth and zenith angle data of each image, and
ee.Terrain.hillShadow algorithm in GEE to identify those pixels with terrain shadows (Zou et al.,
2018; Wang et al., 2020a) (**Fig. 2b**), which were excluded from the data analysis. Out of ~132.43
million pixels in China, approximately 98.36% had more than 5 good-quality observations and
91.24% had more than 10 good-quality observations in 2019. About 93.14% of the 78.9 million
pixels in North China had more than 20 good-quality observations due to the overlapping of
Landsat images at the high latitudes and less cloud cover (Zhou et al., 2019a; Wang et al., 2020b).
Note that number of Landsat-7 ETM+ images in GEE may change in the future, as USGS continues
to work with the International Ground stations (IGS) in the world to assemble and rescue some
images from individual stations. For Landsat-8 OLI images, USGS does not rely on IGS for image
downlink, as its data record is able to store all the images and then downlink them to the Landsat
archive (Wulder et al., 2016).



We used three spectral indices (NDVI, EVI, mNDWI) to identify SWB in this study. These
indices are defined as:
$NDVI = \frac{\rho_{nir}-\rho_{red}}{\rho_{nir}+\rho_{red}}$    (1)
$EVI = 2.5 \times \frac{\rho_{nir}-\rho_{red}}{\rho_{nir}+6\times\rho_{red}-7.5\times\rho_{blue}+1}$    (2)
$mNDWI = \frac{\rho_{green}-\rho_{swir}}{\rho_{green}+\rho_{swir}}$    (3)
where $\rho_{blue}$, $\rho_{green}$, $\rho_{red}$, $\rho_{nir}$, and $\rho_{swir}$ are blue, green, red, near-infrared, and shortwave
infrared bands of Landsat images.

**Insert Fig. 2 here**

**2.2.2 Dam and reservoir datasets**
The GlObal GeOreferenced Database of Dams (GOODD) dataset was released in 2020 and it
lists ~38,000 georeferenced dams as well as derived data on their associated catchments through
one by one degree titles on the Google Earth geobrowser during 2007-2011 and the Shuttle Radar
Topography Mission (SRTM) Water Body Dataset (SWBD) (Mulligan et al., 2020). It provides the
raw digitized coordinates for the locations of dam walls, but it does not provide the detailed
attribute data on the characteristics of each dam and reservoir (**Fig. 3a, d**). Both the large dams
and medium sized dams were captured in this dataset.
The Global Reservoir and Dam (GRanD) Database v1.3 was recently updated in February
2019 by Lehner et al. (2011) (**Fig. 3b, e**). The spatial information of these dams was contributed
by eleven participating institutions. Each dam was assigned to a polygon that depicted the reservoir



surface, which was provided by SWBD (v1.1) and the surface water maps produced by the Joint
Research Center (JRC) of the European Commission from Landsat imagery at 30-m spatial
resolution for the period 1984-2015 (Pekel et al., 2016) (v1.3). All reservoirs with a storage
capacity of more than 0.1 km$^3$ were included in this dataset, and some smaller reservoirs were also
added when their data were available.

The Georeferenced global Dam And Reservoir dataset (GeoDAR) was produced by utilizing

multi-source dam and reservoir inventories (ICOLD WRD and GRanD v1.3 datasets) and the
Google Maps geocoding API (Wang et al., 2021a) (**Fig. 3d, e**). The GeoDAR product includes two
successive versions. GeoDAR v1.0 is essentially a georeferenced subset of ICOLD WRD, and
contains more than 20,000 dam entries, and each of which is indexed by an encrypted identifier
(ID) that is associated with a WRD record, allowing for the potential retrieval of all its 40+
proprietary attributes from ICOLD. GeoDAR v1.1 consists of (1) dam entries as in v1.0 except
those that further harmonized with GRanD for an improved inclusion of the largest dams, and (2)
reservoir boundaries for most of the dam entries.

**Insert Fig. 3 here**

**2.3 Methods**

The workflow for producing the China-LDRL dataset included major two sections: 1)

generation of yearlong SWB maps in China by analyzing time-series Landsat imagery in 2019
with GEE platform, and 2) identification of dams and classification of yearlong SWB into lakes,





reservoirs, and rivers by analyzing the historical satellite images in 2019 within China in Google
Earth Pro. A flowchart showing the methodology of this study is illustrated in **Fig. 4**.
**Insert Fig. 4 here**
**2.3.1 Algorithm to generate annual map of yearlong surface water bodies**

In this study, we combined a surface water index (mNDWI) and two greenness-based

vegetation indices (EVI and NDVI) to identify SWB through the algorithm of ((mNDWI > EVI or
mNDWI > NDVI) and EVI < 0.1) (Eq. (4)). This mNDWI/VIs algorithm can reduce the effects of
vegetation on identification of SWB, and was widely used to identify and map SWB at the regional
and national scales with high accuracy (Zou et al., 2018, 2017; Zhou et al., 2019b; Wang et al.,
2020a). Furthermore, this mNDWI/VIs algorithm had been compared with other surface water
body mapping algorithms (e.g. NDWI, mNDWI, TCW, and AWEI), and the results showed that
this algorithm and Landsat images could identify SWB with high producer's accuracy (98.1%)
and user's accuracy (91.0%) (Zhou et al., 2017).

Surface water body frequency ($F_{SWB}$) of a pixel was calculated as the ratio of the number of

observations identified as surface water body to the number of good-quality observations in a year
and scaled from 0 to 1.0 (or 100%) (Zou et al., 2017), see Eq. (5). We generated the $F_{SWB}$ map
of all the pixels in China for 2019 in the GEE platform (**Fig. 5a**).
$$SWB = \begin{cases} 1, & (\text{mNDWI>EVI or mNDWI>NDVI}) \text{ and EVI<0.1} \\ 0, & \text{Other values} \end{cases} \qquad (4)$$
$$F_{SWB} = \frac{N_{SWB}}{N_{good}} \qquad (5)$$

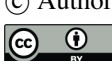



where $SWB$ is surface water body, $F_{SWB}$ is surface water body frequency, $N_{SWB}$ is the number
of observations identified as SWB (see Eq. (4)) in 2019, $N_{good}$ is the number of good-quality
observations in 2019.

Consistent with our previous publications (Zou et al., 2018; Wang et al., 2020a), a water pixel

was defined as yearlong surface water ($F_{SWB} \geq 0.75$), seasonal surface water ($0.05 \leq F_{SWB} <$
$0.75$), or ephemeral surface water ($F_{SWB} < 0.05$). We generated the seasonal and yearlong SWB
maps in China for 2019, respectively (**Fig. 5b, c**).

**Insert Fig. 5 here**

**2.3.2 The procedure to identify dams, reservoirs, and lakes in Google Earth Pro**

We first generated the yearlong SWB vector map in China for 2019 based on the yearlong

SWB raster map, then reprojected it to the Krasovsky_1940_Albers equal-area conic projection
and calculated the area of each yearlong SWB polygon within China (Python code is available in:
https://drive.google.com/drive/folders/1B19VKbCIoDPmu-IcmiZcOIUF8wi1YnE?usp=sharing).
When we reported large reservoirs and lakes, only polygon with area $> 1$ km$^2$ was kept in this
study. In an effort to distinguish riverine or off-channel reservoirs from lakes, we uploaded the
large SWB vector layer into Google Earth Pro, and checked whether a dam existed around each
polygon through the historical satellite images in 2019 within China by visual image interpretation
approach. If a dam did not exist, we classified the polygon as river or lake; if a dam did exist, the
polygon would be classified as impoundment reservoir or off-stream reservoir. Simultaneously, we



classified the corresponding dams as river dams or reservoir dams. Finally, the SWB polygons
were classified into lakes, reservoirs, and rivers, and the dams were classified into reservoir dams
and river dams (**Fig. 4**). This work was carried out and completed by the lead author (Dr. Wang)
over one month, and users could reproduce the dam dataset by uploading the SWB polygons in
the historical satellite images in 2019 in Google Earth Pro and following the procedure described
here. Note that satellite images in the Google Earth Pro may change over time, but such change
may have very limited impact on identification of dam as dam often exists over many years after
its construction.
**2.4 Calculation of lake and reservoir attributes**
The areas (km$^2$) of SWB polygons were generated using the Krasovsky_1940_Albers
coordinate system. As we generated the vector maps of yearlong and seasonal SWB in China for
2019, we calculated the yearlong SWB areas linked to the dams as the reservoir areas. Likewise,
we also calculated each lake area as its attribute in our study.
**2.5 Cross-comparison with other lake and reservoir datasets**
To better understand the improvements and potential application of our China-LDRL dataset,
we compared it with other three available dam and reservoir datasets: the GOODD, GRanD V1.3,
and GeoDAR datasets (**Fig. 3**). We first compared the dam quantity and areas of large reservoir at
the provincial and national scales. Then, we checked the spatial distribution of each dam from
these datasets within Google Earth imagery as all these datasets provide detailed georeferenced



coordinates for some of dams, and the georeferenced information could be directly acquirable from
the spatial longitude and latitude. Here we did not compare the reservoir area with the GOODD
dataset as it does not provide such attribute except catchment area (**Fig. 3d**).
**3. Results**
**3.1 Annual map of surface water bodies in China for 2019**

Surface water body frequency ($F_{SWB}$) of individual pixels for 2019 varied substantially across

China (**Fig. 5**). There were ~3.38 million seasonal surface water pixels (30-m spatial resolution)
in China, amounting to ~3,375.88 × $10^3$ km$^2$ seasonal surface water area (SWA) in 2019. Xinjiang
Province had the largest seasonal SWA (751.14 × $10^3$ km$^2$), followed by Tibet (600.70 × $10^3$ km$^2$),
Qinghai (564.57 × $10^3$ km$^2$), Inner Mongolia (511.42 × $10^3$ km$^2$), and Heilongjiang Province
(343.33 × $10^3$ km$^2$) (**Fig. 6a**). There were ~0.21 million yearlong surface water pixels in China for
2019, amounting to ~214.92 × $10^3$ km$^2$ yearlong SWA, which were mainly located in Tibet (62.65
× $10^3$ km$^2$), Qinghai (41.08 × $10^3$ km$^2$), and Xinjiang (24.60 × $10^3$ km$^2$) Provinces (**Fig. 6b**).
Additionally, Heilongjiang, Jiangsu, Inner Mongolia, Hubei, and Anhui Provinces also had relative
larger yearlong SWA (> 5 × $10^3$ km$^2$) than other provinces in China.

**Insert Fig. 6 here**

**3.2 Numbers and areas of yearlong surface water bodies with different sizes in China**

The numbers and areas of yearlong SWB polygons of different sizes in China differed

considerably for 2019 (**Fig. 7**). In terms of yearlong SWB numbers, out of a total of 3.52 × $10^6$





yearlong SWB polygons in China in 2019, approximate $3.51 \times 10^6$ polygons (99.57%) had an area
of $\leq 1$ km$^2$, and ~$2.16 \times 10^6$ polygons (61.19%) had an area of $\leq 0.0036$ km$^2$ (covering only $2 \times 2$
Landsat grid cells). Only $15 \times 10^3$ (0.43%) yearlong SWB polygons had an area of $> 1$ km$^2$, and
359 polygons had an area of $> 100$ km$^2$. In terms of yearlong SWB areas, out of a total of 214.92
$\times 10^3$ km$^2$ yearlong SWA in China in 2019, large SWB polygons (size $> 1$ km$^2$) accounted for
83.54%, and very large SWB polygons (size $> 100$ km$^2$) accounted for 52.48%.
The numbers and areas of yearlong SWB polygons of different sizes at the provincial scale
had similar distribution patterns with those at the national scale (**Fig. S2, S3**). Almost all the
yearlong SWB polygons in individual provinces had an area of $\leq 1$ km$^2$ (**Fig. S2**), however, those
SWB polygons with an area of $> 1$ km$^2$ accounted for a large proportion of SWA in most provinces
(**Fig. S3**). Those yearlong SWB polygons with an area of $> 100$ km$^2$ were mostly very large lakes
and rivers, and they were mainly located in Tibet, Xinjiang, Qinghai, Jiangxi, and Heilongjiang
Provinces (**Fig. S3**) (Feng et al., 2019). Some provinces also had very large-size reservoirs, such
as Miyun Reservoir in Beijing, whose polygon size was greater than 100 km$^2$.

**Insert Fig. 7 here**

**3.3 Numbers, areas, and distribution of large dams, reservoirs, and lakes in China**
We identified 2,140 large dams in China, including 1,494 reservoir dams and 646 river dams,
most of which were located in South, East, and Northeast China, as well as Tianshan Mountains
in Xinjiang of Northwest China (**Fig. 8a**). At the provincial scale, Heilongjiang Province had the

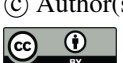



largest number of reservoir dams (148), followed by Shandong (147), Hubei (143), Guangdong
(122), and Jilin (121) Provinces. There were also five provinces (Xinjiang, Yunnan, Liaoning,
Henan, and Anhui) had relatively larger reservoir dam numbers (> 50) than other provinces.
Shanghai (1), Tibet (1), and Qinghai Province (2) had very small numbers of reservoir dams (< 5)
(**Fig. 8b**). Most of river dams in China were distributed in those provinces with large rivers.
Guangdong had the largest number of river dams (90) in China, followed by Sichuan (84), Hunan
(58), Fujian (43), and Yunnan Province (40). However, there were no river dams in Jiangsu and
Shanghai (**Fig. 8c**). In terms of the functions of two kinds of dams and the spatial patterns of
climate (e.g. precipitation, temperature) and social-economic factors (e.g. population, GDP,
irrigation area) in South and North China, the provinces in Northeast and East China had larger
percentage of reservoir dams, whereas the provinces in South and Southwest China had larger
percentage of river dams (**Fig. 8d**).
**Insert Fig. 8 here**
China had 3,508 large lakes with an area of > 1 km$^2$ in 2019, most of which were distributed
in West China, the Lower Yangtze River Basin, and Northeast China (**Fig. 9a, S4**), and they
together amounted to ~75.97 × 10$^3$ km$^2$. Tibet in West China had the largest lake number (978),
followed by Qinghai (482), Xinjiang (388), Inner Mongolia (241), and Hubei Province (218) (**Fig.**
**9b**). The lake areas in China had similar spatial patterns with the lake numbers (**Fig. 9c**), and the
western provinces in China had much larger lake areas than other provinces, especially Tibet and
Qinghai Provinces with 32.51 × 10$^3$ km$^2$ and 16.47 × 10$^3$ km$^2$, respectively. As reservoirs and dams





usually exist simultaneously, the spatial patterns of reservoir numbers and areas matched well with
those of dam numbers (**Figs. 8b, 9e-f**). In total, China had 1,976 large reservoirs in 2019, they
together amounted to an area of ~16.42 × 10$^3$ km$^2$. Hubei Province in Northeast China had the
largest reservoir area (2177.96 km$^2$), followed by Jilin (1,323.29 km$^2$), Heilongjiang (1,320.40
km$^2$), and Henan Province (1304.60 km$^2$). In contrast, Tibet (18.34 km$^2$), Shanghai (36.14 km$^2$),
and Taiwan Province (54.89 km$^2$) had much smaller reservoir areas than other provinces in China.
In general, most of the dams and reservoirs in China were distributed in South China, East China,
and Northeast China, whereas most of lakes were located in West China, the Lower Yangtze River
Basin, and Northeast China (**Figs. 8, 9**).
**Insert Fig. 9 here**
**4. Discussion**
**4.1 Improvements of the dataset of large dams, reservoirs, and lakes in China**
In order to validate the reliability of our China-LDRL dataset, we first compared the numbers
of large dams and areas of large reservoirs between our dataset and published datasets (GOODD,
GRanD, and GeoDAR), then we checked the geographical coordinates of dams within the
historical satellite images in 2019 in Google Earth Pro.
The GOODD dataset has the largest number of dams (9,231) in China among these published
global datasets (**Fig. 10a**). However, it includes both large, moderate, and small dams, and does
not report the corresponding reservoir attributes (e.g. reservoir area), which limits its applications
to water-related research (Paredes-Beltran et al., 2021). The GRanD dataset has the smallest
number (814) of large dams with reservoir area > 1 km$^2$ in China (**Fig. 10b, e**) as the dam
information was provided by multiple institutions from the world (Lehner et al., 2011), which
clearly underestimates the number of dams. The GeoDAR dataset has a larger number of large
dams (993) than the GRanD dataset, because it was generated by combining the GRanD and
ICOLD WRD datasets (Wang et al., 2021a). However, our China-LDRL dataset identified 2,140
large dams and 1,976 large reservoirs (**Fig. 10d, e, f**), making substantial improvement of large
dam and reservoir dataset in China. The number differences of large dams between our China-
LDRL and the GRanD and GeoDAR datasets could be explained by several factors. First, our
study used all the available Landsat images in 2019 and accurate SWB mapping algorithm to
generate SWB maps in China, however, the GRanD and GeoDAR datasets used the SWBD map
(produced in 2000) (Slater et al., 2006) and the surface water maps during 1984-2015 produced by
the JRC (Pekel et al., 2016), thus, we could integrate more Landsat images and get more SWB
polygons, as well as larger numbers of large dams and reservoirs than other datasets. In addition,
the different strategies for identifying dams also caused the differences of dam numbers. The dam
information from the GRanD dataset was contributed by eleven participating institutions, and the
GeoDAR dataset combined two published dam datasets (WRD and GRanD) and rechecked
detailed dam information, then reported the georeferenced information. Unlike the GRanD and
GeoDAR datasets, our study first generated SWB raster and vector maps using the mNDWI/VIs
SWB mapping algorithm, and then selected the large yearlong SWB polygons with area > 1 km$^2$.
After that, we visually checked the large SWB polygons one by one and identified each dam with



accurate geographical coordinates.

**Insert Fig. 10 here**

In addition to the dam numbers, we also compared the reservoir areas between different

datasets (**Fig. 11**). Our China-LDRL dataset reports ~16.42 × 10$^3$ km$^2$ large reservoir area, which
was smaller than those of the GRanD (20.98 × 10$^3$ km$^2$) and GeoDAR (21.84 × 10$^3$ km$^2$) datasets.
We checked the reservoir polygons of the three datasets in Google Earth Pro, and found that some
large lakes were identified as reservoirs by the GranD and GeoDAR datasets, such as the Hongze
Lake in Jiangsu Province (**Fig. S5a**), contributing to the overestimate of reservoir areas. In addition,
the GRanD v1.3 dataset linked the "maximum surface water extent" from the JRC dataset to the
corresponding dams as the reservoir regions, however, we used the "yearlong surface water body"
to depict the reservoir in the China-LDRL dataset, which caused our smaller reservoir areas (**Fig.
S5b-e**).

**Insert Fig. 11 here**

In this study, we also checked the accuracy of geographical coordinates of dams from these

dam datasets. Here we first uploaded above-mentioned three dam datasets and our China-LDRL
dataset in the Google Earth Pro and visually checked the spatial distribution of each dam within
the historical satellite images in 2019 (**Fig. 12**). We found that the dam locations of the GOODD
dataset had substantial geographic offsets, some of which are larger than 500 m (**Fig. S6**). We
further overlapped the GOODD dam layer with our yearlong SWB map (Section 2.3.1), and the





results showed that only 12.52 ± 3.87% of the GOODD dams were intersected with the SWB layer
at the national scale (**Fig. 13a**). In the case that we applied a 100-m and 500-m tolerance when
intersecting the GOODD dams with our yearlong SWB map for 2019, the intersection rate
increased to only 47.58 ± 9.70% and 76.46 ± 7.11%, respectively (**Figs. 13b, S7**). In addition, we
applied different tolerances when intersecting the GRanD and GeoDAR datasets with our yearlong
SWB layer. About 65.57 ± 6.79% of the dams in the GRanD dataset were intersected with our
yearlong SWB map (**Fig. 13a**), which increased to 87.52 ± 6.45% and 95.94 ± 4.49% when using
a 100-m and 500-m tolerance (**Figs. 13b, S7**). Although the GeoDAR dataset is released by
integrating the GRanD dataset, its geographical coordinates also had larger offsets (**Fig. 12d, f, g**),
and 41.10 ± 6.13% of its dams were intersected with the yearlong SWB layer, and 63.18 ± 5.61%
and 86.69 ± 3.74% intersected when the tolerance was 100-m and 500-m (**Figs. 13b, S7**). These
comparisons suggested the substantial geographic offsets of these published datasets (GOODD,
GRanD, and GeoDAR), and improved accuracy of our China-LDRL dataset, which could provide
important and reliable information for water resource management and water security in China.

**Insert Fig. 12 here**

**Insert Fig. 13 here**

**4.2 Uncertainties, limitations, outlooks, and implications**

In this study, we produced detailed and accurate China's open surface water bodies, large

dams, reservoirs, and lakes (China-LDRL) dataset for 2019, and analyzed their spatial distribution



patterns. This study benefited from the usage of time-series Landsat imagery and GEE cloud
computing platform, as well as simple and robust SWB mapping algorithms. First, time series
Landsat images at high spatial resolution (30-m) provide larger numbers of good-quality
observations for identifying SWB. Second, GEE cloud computing platform enables us to acquire
and analyze tens of thousands of Landsat images in hours. Third, the mNDWI/VIs algorithm used
in this study could reduce the uncertainties induced by the bad-quality observations and provide
accurate SWB maps. Finally, we visually checked the large SWB polygons (area > 1 km$^2$) one by
one by using the historical satellite images in 2019 within China in Google Earth Pro, and we
recorded the georeferenced coordinates of individual dams in China for 2019.
We would also acknowledge that the data quality of input satellite images remains to be a
concern for the identification of dams, reservoirs, and lakes. The spatial distribution of good-
quality observations of Landsat data shows that more than 98.36% of the total 30-m pixels in China
had more than 5 good-quality observations and more than 91.24% of the total pixels had more than
10 good-quality observations for 2019 (**Fig. 2b**), but the regions with complex topography and
mountains, such as South and Southwest China, had much fewer good-quality observations than
other regions, which might underestimate surface water areas, as well as dam and reservoir
numbers and areas. In addition, it is impossible to remove all the bad-quality observations (e.g.
clouds, terrain shadows) because of the limited quality of the QA band and digital elevation model
data in GEE. Therefore, the remaining bad-quality observations could result in some inevitable
uncertainties in the resultant maps. In the future, as more images from Landsat dataset and other



high spatial resolution sensors (e.g., Sentinel-1, Sentinel-2) are added into GEE platform (Wulder
et al., 2016), SWB mapping accuracy could be further improved, providing more detailed
geospatial data of dams, reservoirs, and lakes in China.

In our China-LDRL dataset, we identified and reported those large SWB, however, the

importance of monitoring small water bodies (area $\leq 1$ km$^2$) and dams is gradually recognized as
they play critical roles in accurate assessments of their agricultural potential or their cumulative
influence in watershed hydrology (Ogilvie et al., 2018). In the near future, we can include these
small SWB polygons into our dataset to enhance the spatial details and distributions of dams,
reservoirs, and lakes in China.

The conversions between rivers, lakes, and reservoirs have critical effects on the ecosystem

services. For example, the construction of the Three Gorges Dam contributed to the decrease of
surface water area and biodiversity in its downstream areas (Fang et al., 2006; Feng et al., 2013;
Wang et al., 2020a), and reduced the sediment loads in the Yangtze River, causing the decreased
deposition rates of coastal wetlands in the Yangtze Delta (Feng et al., 2016; Wang et al., 2021b).
Furthermore, the conversion from natural lakes and rivers to man-made reservoirs has
disproportionate effects on the local, regional, and global carbon cycle (Howard Coker et al., 2009).
For example, dam construction has reduced the areal extent of $CO_2$ gas exchange in natural rivers
(Ran et al., 2021). In the future, more detailed information (e.g. construction year of dam) could
be included in our China-LDRL dataset, making it possible to analyze the effects of conversions
from natural lakes and rivers to reservoirs on the biodiversity and carbon cycle.




## 5. Data availability

The China-LDRL dataset is publicly available at
https://doi.org/10.6084/m9.figshare.16964656.v2 (Wang et al., 2022), and it includes three
shapefiles. The "China_large_dams_attribute.shp" is the large dams in China, as well as their
attributes (including ID, dam class, longitude and latitude, polygon area, and corresponding
reservoir ID). The "China_large_lakes.shp" and "China_large_reservoirs.shp" are the large lakes
and reservoir maps in China.

## 6. Code availability

Code used in calculations of surface water bodies is available upon request. Code for transferring
the images to vector maps in Python could be found in:
https://drive.google.com/drive/folders/1B19VKbCIoDPmu-IcmiZcOIUF8wi1YnE?usp=sharing.

## 7. Conclusion

Several studies have published global or national dam, reservoir, and lake datasets based on
satellite images (**Table 1**). However, these datasets usually have large georeferenced coordinate
offsets, which poses some limitations to those studies that aim to address major issues in hydrology,
ecology, and water resource management in China. In this study, we generated the dataset of





China's open surface water bodies, large dams, reservoirs, and lakes (China-LDRL) for 2019, and
then analyzed their spatial distributions at the provincial and national scales. Satellite image data
quality is still a major source of uncertainty that affects the accuracy of the surface water body
maps. As more images from Landsat datasets and other high spatial resolution sensors (e.g.,
Sentinel-1, Sentinel-2) are added into GEE platform, the accuracy of SWB maps can be further
improved, providing more detailed geospatial data of dams, reservoir, and lakes in China. The
provision of the reliable, accurate China-LDRL dataset on dams, reservoirs, and lakes will
contribute to the understanding of water crisis and water resources management in China.
**Author contribution**
X.X., X.W., and B.L. designed the study. X.W. carried out image data processing and led
interpretation of the results and writing of the manuscript. Y.Q., and J.D. contributed to image data
processing, X.X., B.L., Y.Q., J.D., and J. W. contributed to the interpretation and discussion of the
results.
**Declaration of Competing Interest**
The authors declare that they have no known competing financial interests or personal
relationships that could have appeared to influence the work reported in this paper.
**Acknowledgements**
This study was supported in part by research grants from the U.S. National Science Foundation
(1911955), the Natural Science Foundation of China (81961128002), the China Postdoctoral



Science Foundation (2021M700835 and 2021TQ0072), and the China Scholarship Council

465    (201906100124).

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

**Figures and figure legends**

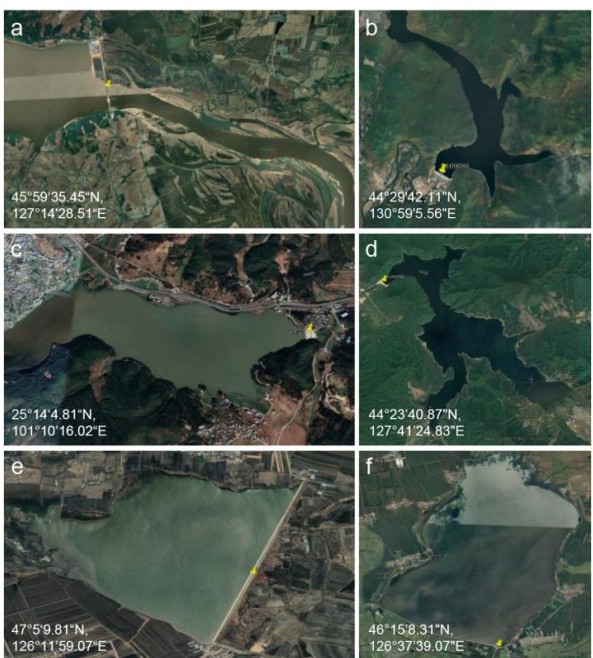


**Fig. 1.** Types of reservoirs in China within high-resolution images (© Google Earth Pro 2019). (a-
b) Impoundment reservoir; (c-d) Mountain reservoir; (e-f) Plain reservoir.

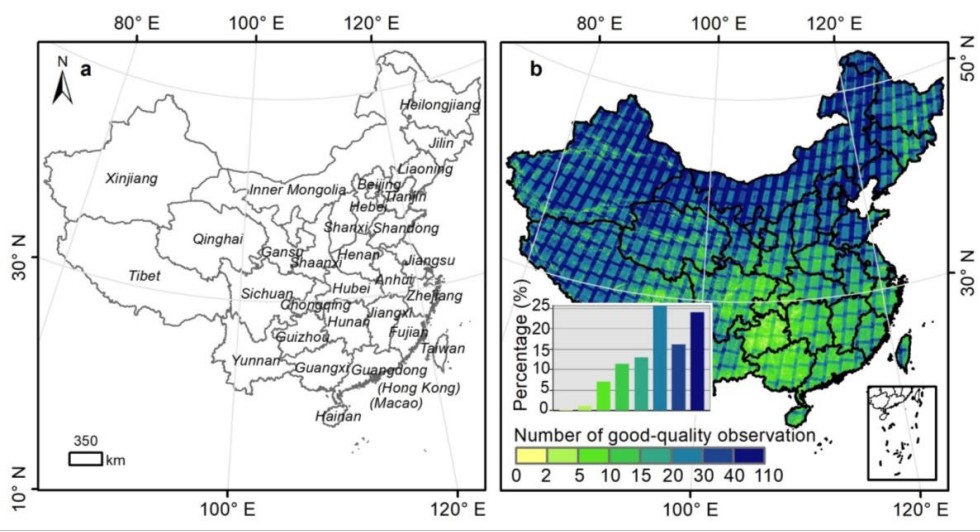

**Fig. 2.** Spatial distribution of provinces (a) and numbers of Landsat good-quality observations (b)

in China for 2019.

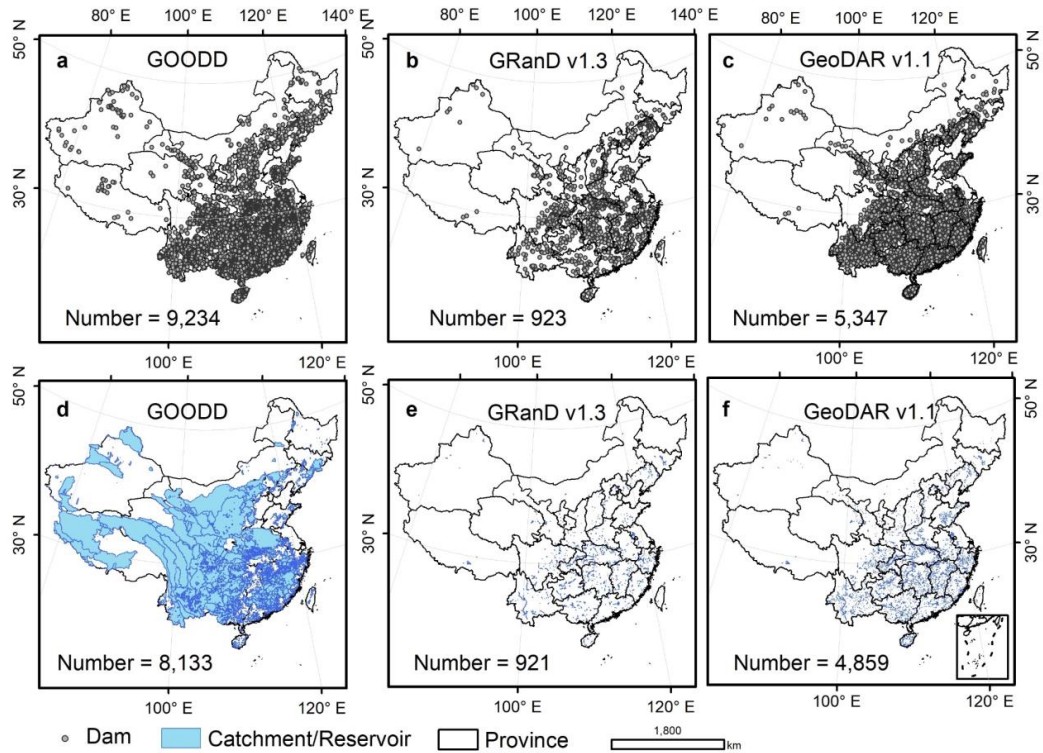

**Fig. 3.** Spatial distribution of dams from the GlObal GeOreferenced Database of Dams (GOODD)

(a) (Mulligan et al., 2020), the Global Reservoir and Dam (GRanD) v1.3 (b) (Lehner et al., 2011),

and the Georeferenced global Dam And Reservoir (GeoDAR) v1.1 (c) (Wang et al., 2021) datasets.

The GOODD dataset reported the catchment of each dam (d) while the GRanD and GeoDAR

datasets reported the reservoir information of each dam (e, f).



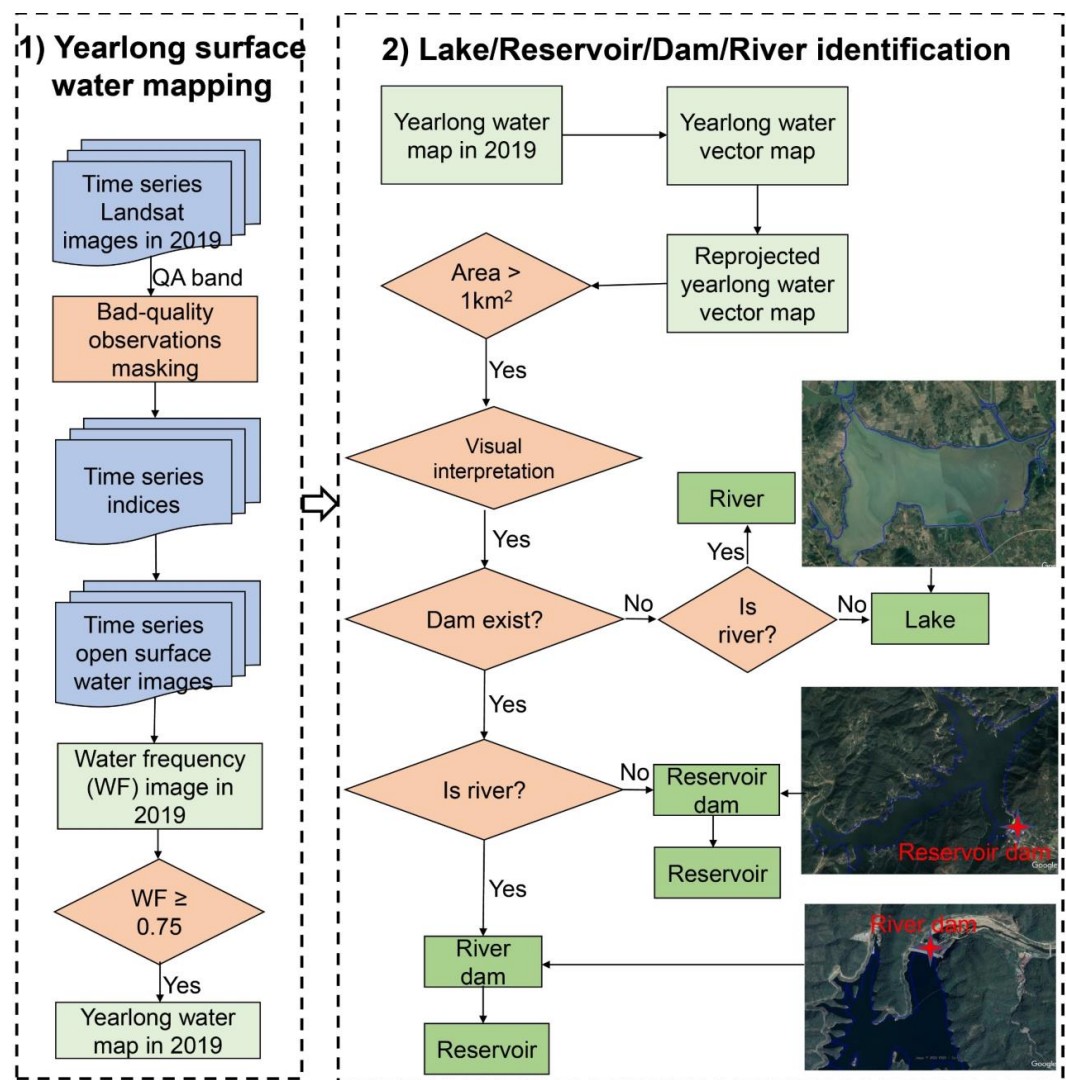


**Fig. 4.** Schematic flowchart of lakes, reservoirs, dams, and rivers identification in this study. The

images were acquired from Google Earth Pro (© Google Earth Pro 2019).

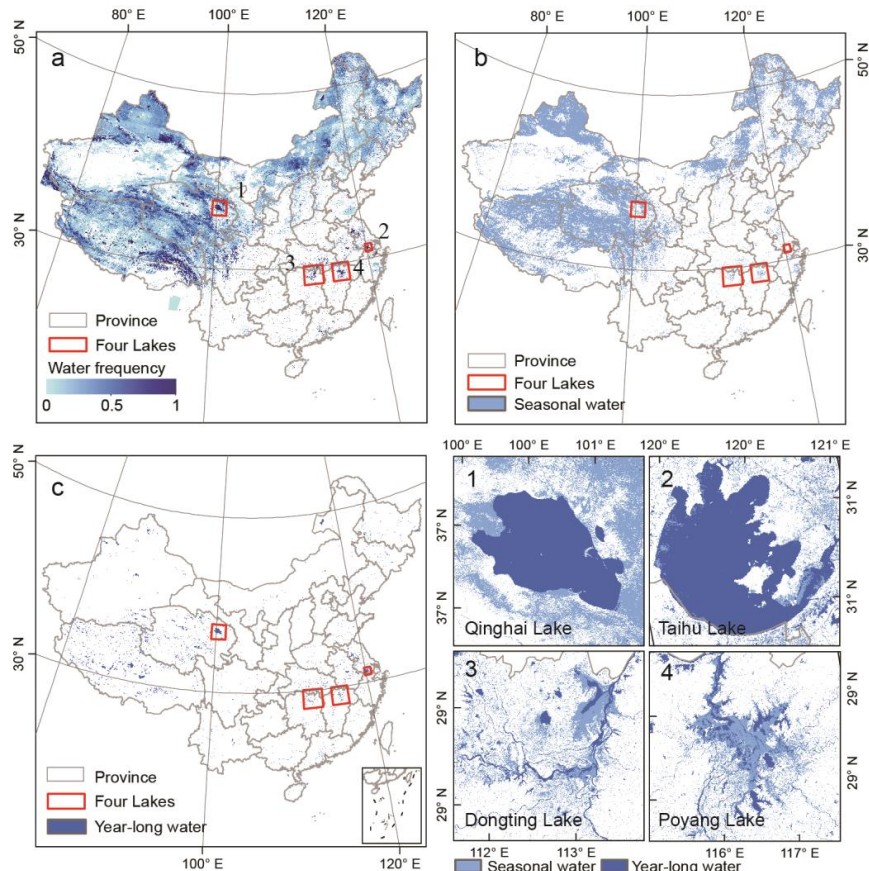


**Fig. 5.** Spatial distribution of surface water body (SWB) in China for 2019. (a), Water frequency,

(b), Seasonal SWB, (c), Yearlong SWB. Subfigures (1-4) are three zoom-in views of seasonal and

year-long SWB in Qinghai Lake, Taihu Lake, Dongting Lake, and Poyang Lake in China.

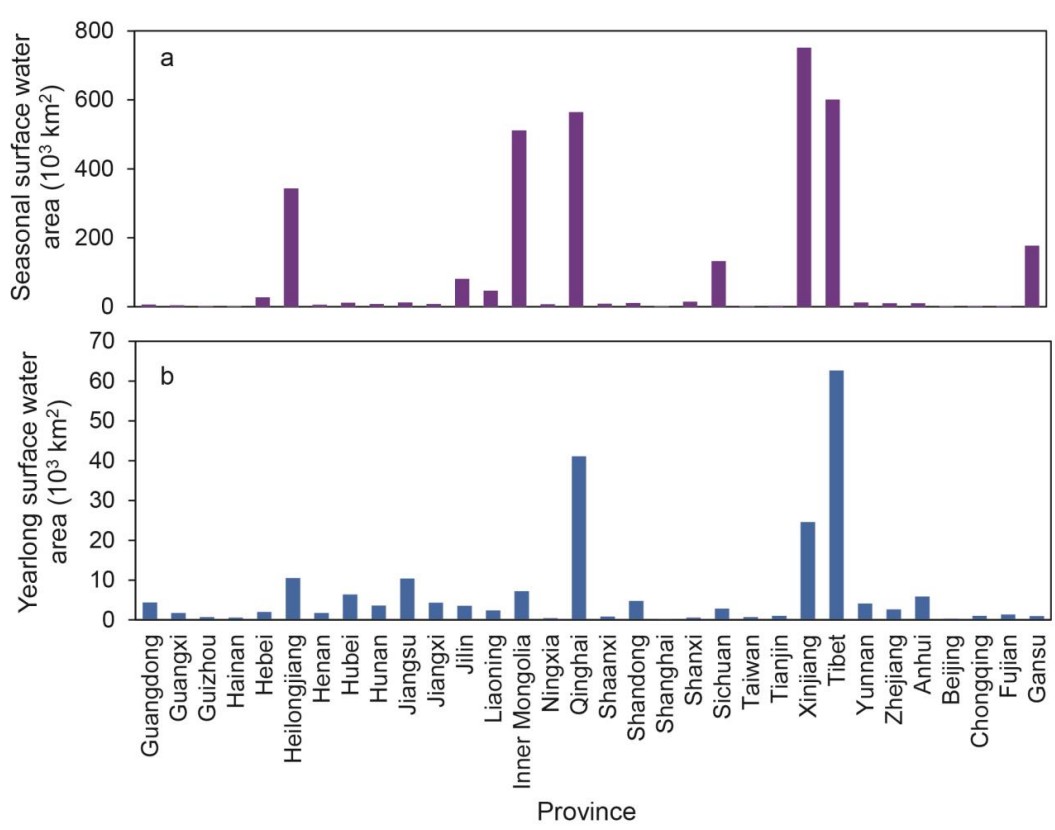


**Fig. 6.** Areas of seasonal (a) and yearlong (b) surface water bodies by province in China for 2019.

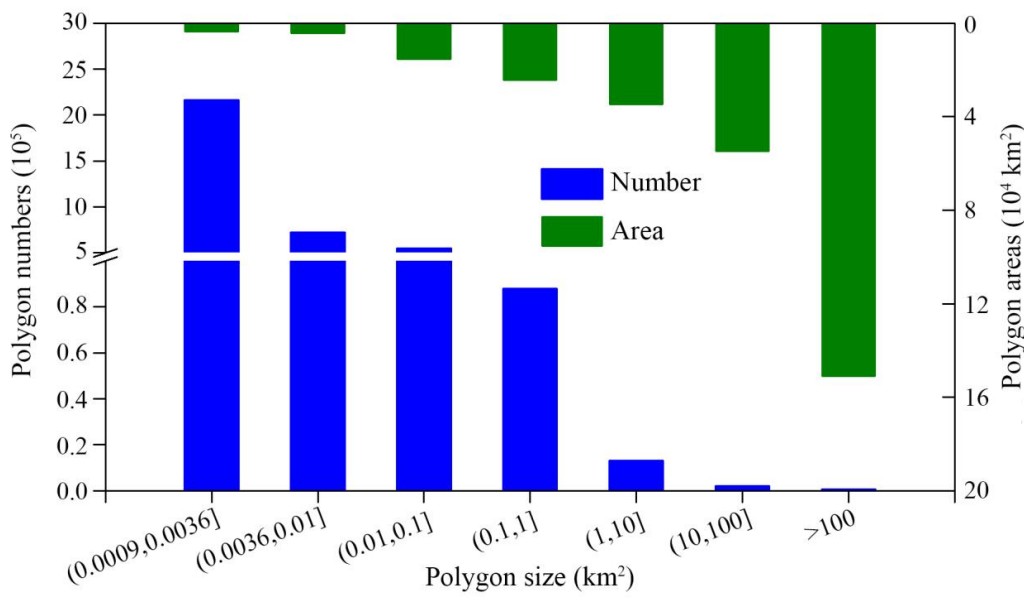


**Fig. 7.** Numbers and areas of yearlong surface water body polygons with different sizes.

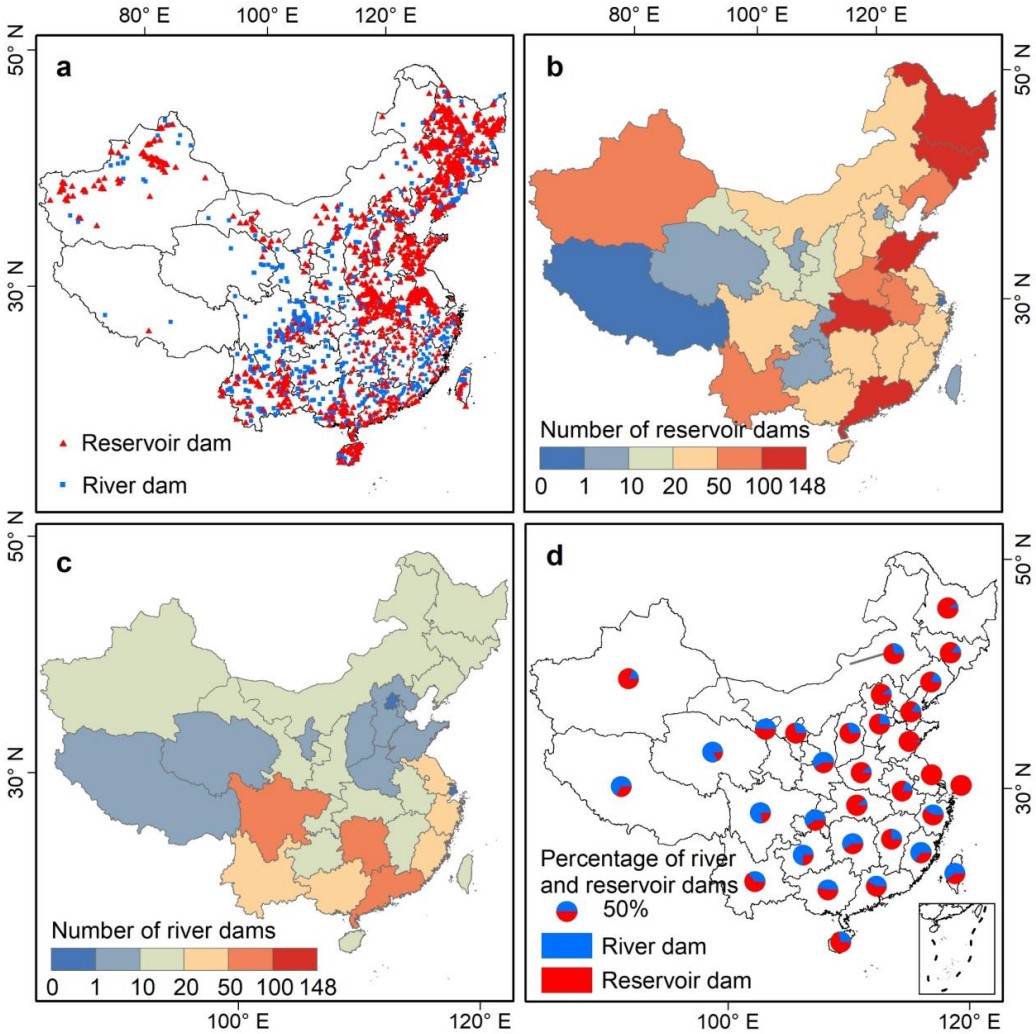


**Fig. 8.** Distribution of river dams and reservoir dams in China for 2019. **a**, Spatial distribution of

river dams and reservoir dams; **b**, Number of reservoir dams by province; **c**, Number of river dams

by province; **d**, Percentage of river dams and reservoir dams by province.

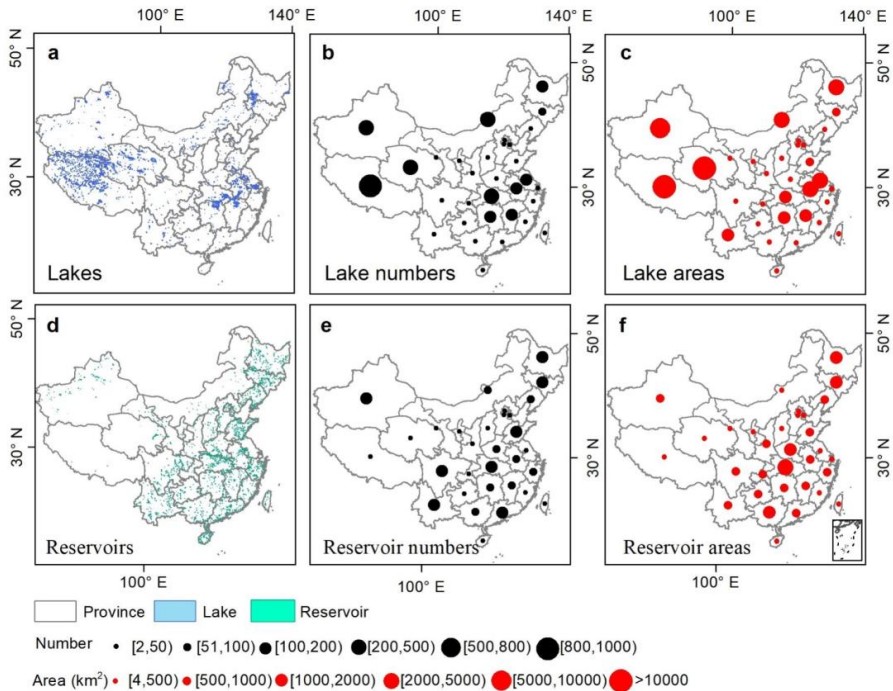


**Fig. 9.** Distribution of lakes and reservoirs in China. **a**, Spatial distribution of lakes; **b**, Lake

numbers by province; **c**, Lake areas by province; **d**, Spatial distribution of reservoirs; **e**, Reservoir

numbers by province; **f**, Reservoir areas by province.



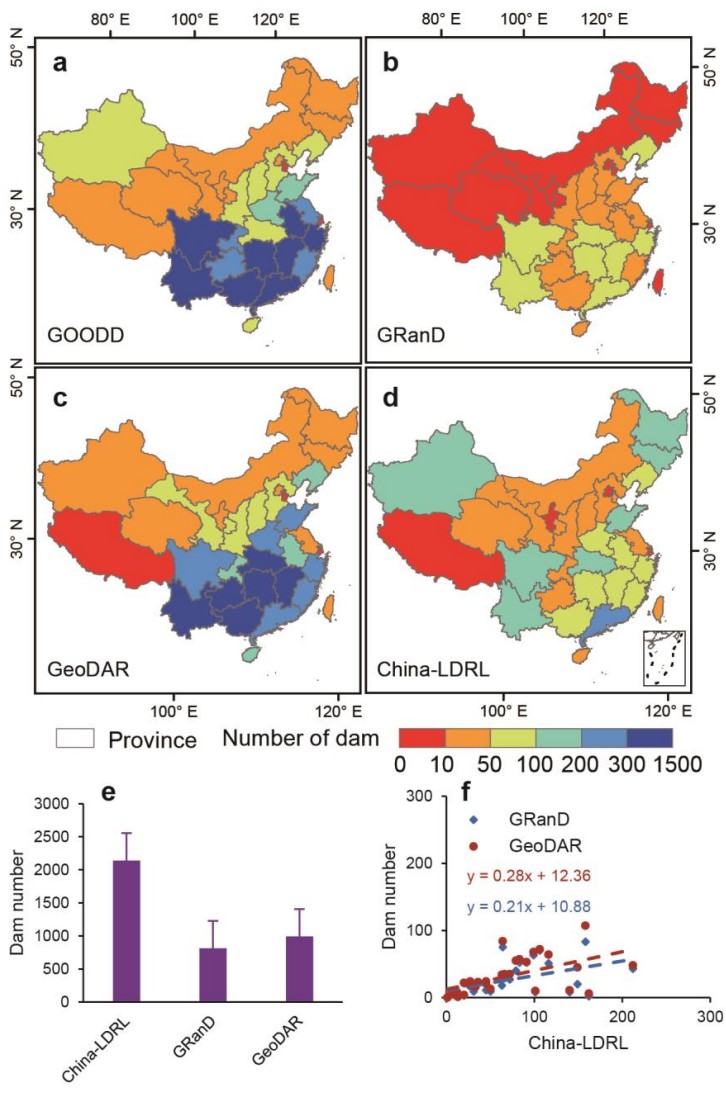

**Fig. 10.** Numbers of large dams of different datasets. (a) Dam number in the GOODD dataset by

province; (b) Dam number in the GRanD dataset by province; (c) Dam number in the GeoDAR

dataset by province; (d) Dam number in the China-LDRL dataset by province; (e) Large dam

numbers of different datasets in China; (f) The relationships of large dam numbers between China-

LDRL and GRand and GeoDAR datasets.

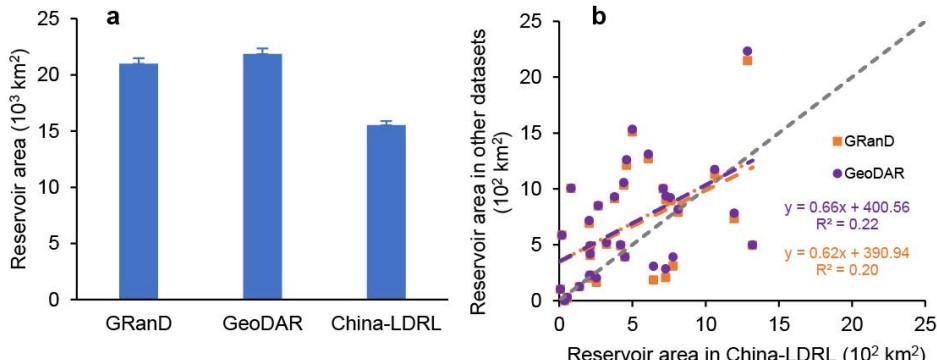


**Fig. 11.** Areas of large reservoir (a) and their relationships (b) of the GRanD, GeoDAR, and our

China-LDRL datasets.





**Fig. 12.** Dam from the GOODD, GeoDAR, and China-LDRL datasets within Google Earth Pro (©

Google Earth Pro 2019).




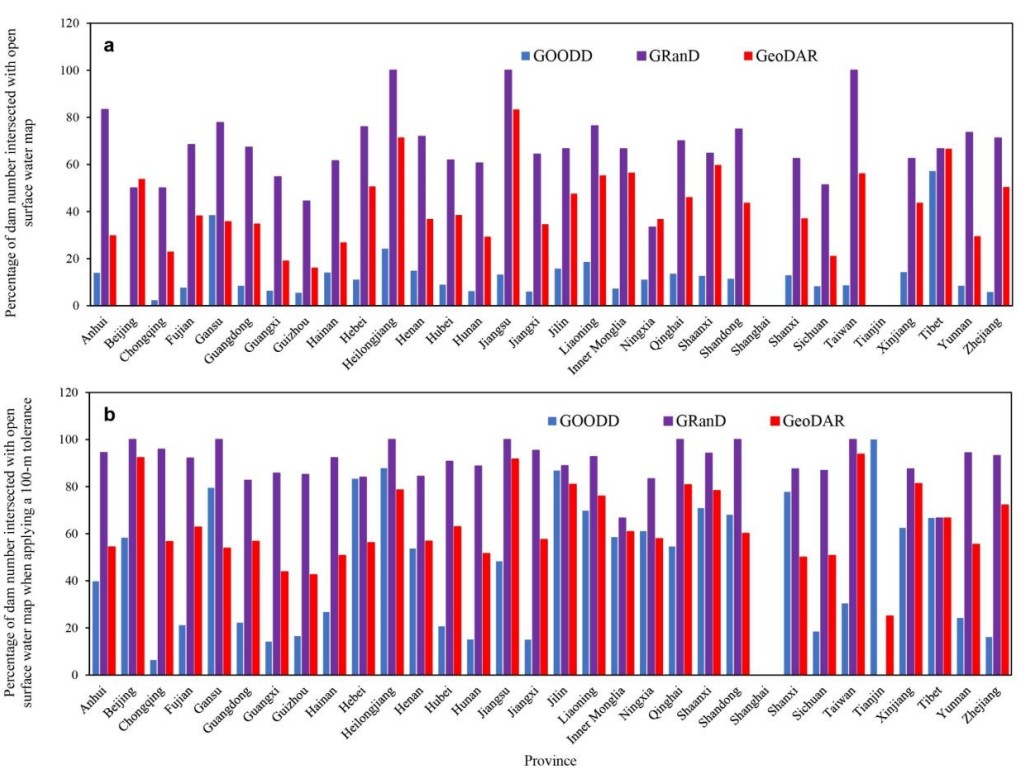


**Fig. 13.** The numbers of dams intersected with surface water body (SWB) map in China for 2019

by province. (a) Percentage of numbers of dams intersected with SWB map; (b) Percentage of

numbers of dams intersected with SWB map when applying a 100-m tolerance.