# Peer review of "Improved maps of surface water bodies, large dams, reservoirs, and"

_Earth System Science Data, 2021_

## Author Response (AR1)

**Response Letter to the Referees**

Here we provided detailed responses to each of the comments from individual referees. For easy reading, we use blue color text for the response from us. Please note that the line numbers in the response letter refer to the revised main text (clear version). The revised China-LDRL dataset is publicly available at https://doi.org/10.6084/m9.figshare.16964656.v3.

**Referee #1:**

In this study, the authors generated a valuable dataset of Large Dams, Reservoirs, and Lakes across China by analyzing all available Landsat imagery in 2019 via GEE platform.

Overall, this manuscript is written well and suitable to publish in ESSD. I recommend a minor revision based on the comments below to improve the quality of this manuscript /data set before publication.

**Reply:** We were encouraged by your positive statement, and we appreciate your time and effort during the review process of our manuscript.

Specific comments:

1) Line 78: Please confirm the expression of "GlObal geOreferenced Database of Dams (GOODD) V1", which is different from the handwriting in line 165.

**Reply:** We double-checked the name of dataset, and its full name is "GlObal GeOreferenced Database of Dams". We have updated it throughout the main text.

2) Line 118: Why did the authors choose to analyze the large SWB with area > 1km$^2$? What about the area ≤ 1km$^2$? Please add one or two sentences to explain it.

**Reply:** Large water bodies (e.g. lakes and reservoirs) have been the subject of great interest not only because of their water resources roles but also as indicators of anthropogenic impact on water scape change (Yang and Lu, 2014). A number of studies have reported and analyzed the dynamics of large water bodies, for example, Feng et al. (2019) explored the overall characteristics and changes of large SWB during 1984-2015 in China; Zhang et al. (2019) explored the evolution of large lake across China during 1960s-2015. Furthermore, Yang and Lu (2014) analyzed the drastic change in China's large lakes and reservoirs over the past decades.

Thus, in our study, we analyzed the large SWB with area > 1km$^2$ and those SWB with area ≤ 1km$^2$ were excluded. In the main text, we introduced the importance of large SWB, including large lakes and reservoirs, in the control and management of water resources: "*Surface water bodies (SWB), including large lakes and reservoirs (surface water areas > 1 km$^2$), play an important role in the control and management of water resources (Yang and Lu, 2014, 2013; Feng et al., 2013, 2019)*" and "*Nearly 50% of the global large dams were built primarily for agricultural irrigation through storing,*"

*regulating, and diverting water (Mulligan et al., 2020). Additionally, they are also used for hydropower generation, human and industrial uses, and flood peak attenuation (Lehner et al., 2011; Lehner and Döll, 2004; Wang et al., 2021a). Large lakes have been the subject of great interest not only because of their water resources but also as indicators of local climate change and anthropogenic activities (Zhang et al., 2019; Ma et al., 2011; Birkett and Mason, 1995), and they could provide vital ecosystem services for human being, such as alteration of river flow, supplies of irrigation water, fisheries, and abundant valuable mineral deposits, and have disproportionate effects on the global carbon cycle (Ran et al., 2021; Armstrong, 2010; Ma et al., 2011).*" (Line 37-51). In addition, we explained the exclusion of small SWB in the Objective of this study: "*The objective of this study was to produce detailed and accurate maps of open SWB, large dams, reservoirs, and lakes (surface water area > 1 km$^2$) in China in 2019, the latest year when this study started in late 2020, and those SWB with area $\leq$ 1km$^2$ were excluded.*" (Line 114-116).

References:

Feng, S., Liu, S., Huang, Z., Jing, L., Zhao, M., Peng, X., Yan, W., Wu, Y., Lv, Y., Smith, A. R., McDonald, M. A., Patil, S. D., Sarkissian, A. J., Shi, Z., Xia, J., and Ogbodo, U. S.: Inland water bodies in China: Features discovered in the long-term satellite data, Proc. Natl. Acad. Sci., 116, 25491-25496, https://doi.org/10.1073/pnas.1910872116, 2019.

Yang, X. and Lu, X.: Drastic change in China's lakes and reservoirs over the past decades, Sci. Rep., 4, 6041, https://doi.org/10.1038/srep06041, 2014.

Zhang, G., Yao, T., Chen, W., Zheng, G., Shum, C. K., Yang, K., Piao, S., Sheng, Y., Yi, S., Li, J., Oreilly, C., Qi, S., Shen, S., Zhang, H., and Jia, Y.: Regional differences of lake evolution across China during 1960s-2015 and its natural and anthropogenic causes, Remote Sens. Environ., 221, 386-404, https://doi.org/10.1016/j.rse.2018.11.038, 2019.

3) Line 127: In the Fig.2a, the DEM or other data could be added as the base map to make this figure look more abundant.

**Reply:** Good suggestion. We added the DEM data of China in Fig. 2a, and reorganized the Fig. 2b to avoid the coverage by the histogram.

[Figure]

**Fig. 2** Spatial distribution of provinces and elevation (a) and numbers of Landsat good-quality observations (b) in China for 2019.

4) Lines 127-132: The content of section 2.1 is a little simple, some information could be added in this part, like geographic or climatic characteristics.

**Reply:** Done. We added some brief introduction about the geographic or climatic characteristics of China in section 2.1: "*China has great altitude diversity as the eastern plains and southern coasts consist of lowlands and foothills, the southern areas of China consist of hilly and mountainous terrains, the west and north of the country are dominated by basins, plateaus, and massifs, and the southwestern China contains part of the highest tablelands on earth, the Tibetan Plateau (Fig. 1a). Due to substantial differences in latitude, longitude, and altitude, the climate of China is extremely diverse, ranging from tropical in the far south to subarctic in the far north and alpine in the higher elevations of the Tibetan Plateau, contributing to the much more surface water areas in Southwest and Southeast of China than other regions, especially North China (Wang et al., 2020a).*" (Line 136-143).

5) Line 146: In the Fig.2b, part of the background map was covered by the histogram, which could be improved to be more normative and beautiful.

**Reply:** Done. We updated this figure and is shown in Fig. 2.

6) Line 170: In the Fig.3, if it's possible, the shapefile of dam and reservoir could be symbolized separately.

**Reply:** In the Fig.3, we showed the spatial distribution of dams and reservoirs from the GOODD, GRanD, and GeoDAR datasets using different symbolizations. Fig.3a-c were the three dam maps using point symbol, and Fig. 3d showed the catchment of each dam of GOODD as it only reported the catchment of each dam rather than the reservoirs. Fig. 3e-f showed the spatial distribution of reservoir from GRanD and GeoDAR.

[Figure]

**Fig.3** Spatial distribution of dams from the GlObal GeOreferenced Database of Dams (GOODD) (a), the Global Reservoir and Dam (GRanD) v1.3 (b), and the Georeferenced global Dam And Reservoir (GeoDAR) v1.1 (c) datasets. The GOODD dataset reported the catchment of each dam (d) while the GRanD and GeoDAR datasets reported the reservoir information of each dam (e, f).

7) Lines 209-210: As we know, the Landsat 7 ETM+ has problem of stripe, would it effect the generation of SWB? How did the authors deal with this issue? Please give detailed explanation.

**Reply:** You raised a good question about the stripe of Landsat-7 imagery. In this study, we used the Landsat surface reflectance (SR) images in the GEE platform, and all these images had undergone necessary pre-processing, including radiometric calibration and atmospheric correction. In addition, all the pixels of the stripes in the Landsat-7 have been removed by GEE team (**Fig. R1**), thus, they were not included in our study when we generated the annual water frequency map of China.

[Figure]

**Fig. R1** The pixel values of the stripe in Landsat-7 SR imagery in GEE.

8) Line 221: In the section 2.3.2, when the authors generated the polygons of lakes or reservoirs, how did the authors determine the borderlines between water bodies and adjacent land, especially for mixed land-water pixels? Please give detailed explanation.

**Reply:** In our study, we first identified SWB through the algorithm of ((mNDWI > EVI or mNDWI > NDVI) and EVI < 0.1), which reduces the effects of mixed land-water pixels on identification of SWB, especially those vegetation pixels, and was widely used to identify and map SWB at the regional and national scales. Second, we generated the surface water frequency map in 2019 using all the available Landsat imagery and the algorithms in Section 2.3.1. Third, the yearlong surface water imagery was generated using water frequency ≥ 0.75. After that, the vector map of yearlong surface water for 2019 was generated based on the yearlong surface water imagery, and was reprojected the Krasovsky_1940_Albers equal-area conic projection and calculated the area of each yearlong surface water polygon within China. Finally, we removed those polygons with area ≤ 1km$^2$ and only reported large reservoirs and lakes with area > 1 km$^2$ in this study (**Fig. R2**). Therefore, the boundaries of resultant lakes and reservoirs were detected automatically. In order to introduced the work flow for generating lake and reservoir polygons clearly, we added the **Fig. R2** in the supplementary file and improved the introduction of Section 2.3.2.

[Figure]

Fig. R2 The schematic diagram of generating the polygons of lakes or reservoirs from water frequency map in the Hongze Lake of Jiangsu Province and Qinghai Lake of Qinghai Province.

9) Lines 240-244: In the section 2.4, is there only "area" in the attribute table? If not, please give more description about the attributes, otherwise, this paragraph is too simple.

**Reply:** Thanks. Here we reported the area and perimeter of lakes and reservoirs, the ID of corresponding dams and dam/reservoir classes of reservoirs as their attributes (**Fig. R3**). In this revised version, we combined section 2.3.2 and 2.4 into one section and introduced the methods of calculating area and perimeter of each polygon. The introduction of each column of the attributes were shown in the Section of "5. Data availability".

**Attributes of lakes**

| FID | Shape * | ID | poly_area | poly_len |
|---|---|---|---|---|
| 0 | Polygon | 0 | 2.561173 | 16.766622 |
| 1 | Polygon | 1 | 2.046933 | 9.651268 |
| 2 | Polygon | 2 | 2.219894 | 8.370396 |
| 3 | Polygon | 3 | 13.330212 | 55.676937 |
| 4 | Polygon | 4 | 38.10306 | 268.454778 |
| 5 | Polygon | 5 | 35.087878 | 110.436371 |
| 6 | Polygon | 6 | 1.479911 | 25.264502 |
| 7 | Polygon | 7 | 6.220322 | 32.306757 |
| 8 | Polygon | 8 | 20.269402 | 90.108679 |
| 9 | Polygon | 9 | 7.880843 | 50.42397 |
| 10 | Polygon | 10 | 1.128003 | 6.014507 |
| 11 | Polygon | 11 | 1.700841 | 8.177997 |
| 12 | Polygon | 12 | 1.727262 | 11.270981 |
| 13 | Polygon | 13 | 1.273662 | 9.44156 |
| 14 | Polygon | 14 | 1.048149 | 8.174765 |
| 15 | Polygon | 15 | 12.686775 | 43.651212 |
| 16 | Polygon | 16 | 7.503791 | 57.605343 |
| 17 | Polygon | 17 | 81.917412 | 215.488834 |
| 18 | Polygon | 18 | 1.060152 | 4.778265 |
| 19 | Polygon | 19 | 1.379701 | 15.202636 |
| 20 | Polygon | 20 | 1.613516 | 9.894215 |
| 21 | Polygon | 21 | 2.622165 | 13.698355 |
| 22 | Polygon | 22 | 640.037245 | 1457.645367 |
| 23 | Polygon | 23 | 3.573744 | 14.875817 |
| 24 | Polygon | 24 | 2.857957 | 13.760246 |

**Attributes of reservoirs**

| FID | Shape * | ID | poly_area | poly_len | dam_ID | dam_class |
|---|---|---|---|---|---|---|
| 0 | Polygon | 0 | 3.338023 | 20.358279 | 56 | -1 |
| 1 | Polygon | 1 | 2.642188 | 11.967802 | 41 | -1 |
| 2 | Polygon | 2 | 2.345197 | 16.788014 | 40 | 1 |
| 3 | Polygon | 3 | 1.727991 | 6.432137 | 45 | -1 |
| 4 | Polygon | 4 | 2.481339 | 16.527029 | 39 | 1 |
| 5 | Polygon | 5 | 1.685698 | 8.485414 | 16 | 1 |
| 6 | Polygon | 6 | 1.505444 | 11.876958 | 57 | 1 |
| 7 | Polygon | 7 | 7.131356 | 27.404409 | 44 | 1 |
| 8 | Polygon | 8 | 1.985066 | 7.905328 | 42 | 1 |
| 9 | Polygon | 9 | 3.280292 | 42.853447 | 15 | 1 |
| 10 | Polygon | 10 | 3.951754 | 21.413478 | 43 | -1 |
| 11 | Polygon | 11 | 2.441946 | 23.850471 | 14 | 1 |
| 12 | Polygon | 12 | 1.797849 | 10.512701 | 46 | -1 |
| 13 | Polygon | 13 | 2.746095 | 10.981722 | 47 | 1 |
| 14 | Polygon | 14 | 2.745012 | 18.481081 | 37 | 1 |
| 15 | Polygon | 15 | 3.98678 | 27.401023 | 17 | 1 |
| 16 | Polygon | 16 | 1.10316 | 8.293268 | 38 | -1 |
| 17 | Polygon | 17 | 1.554771 | 9.160939 | 58 | 1 |
| 18 | Polygon | 18 | 3.555699 | 13.140496 | 49 | 1 |
| 19 | Polygon | 19 | 2.960094 | 14.982563 | 48 | 1 |
| 20 | Polygon | 20 | 2.685784 | 8.564417 | 50 | 1 |
| 21 | Polygon | 21 | 1.816469 | 7.84277 | 59 | 1 |
| 22 | Polygon | 22 | 2.21384 | 11.901032 | 55 | -1 |
| 23 | Polygon | 23 | 1.300648 | 7.562165 | 13 | 1 |
| 24 | Polygon | 24 | 1.186143 | 8.571778 | 51 | 1 |
| 25 | Polygon | 25 | 3.025715 | 18.126664 | 52 | 1 |
| 26 | Polygon | 26 | 1.80524 | 13.752735 | 19 | 1 |
| 27 | Polygon | 27 | 1.308694 | 7.887077 | 18 | 1 |
| 28 | Polygon | 28 | 1.193683 | 7.922579 | 54 | -1 |
| 29 | Polygon | 29 | 3.074667 | 13.11458 | 60 | 1 |
| 30 | Polygon | 30 | 6.121569 | 43.582946 | 20 | 1 |
| 31 | Polygon | 31 | 2.225887 | 10.124161 | 53 | 1 |
| 32 | Polygon | 32 | 1.100876 | 6.320478 | 61 | -1 |
| 33 | Polygon | 33 | 4.56658 | 21.125573 | 62 | -1 |

Fig. R3. Attributes of lakes and reservoirs in our dataset.

**Referee #2:**

In this manuscript, Wang et al. produced a database entitled China-LDRL, which contains (1) thousands of detected large (>1 km$^2$) permanent water bodies (excluding free-flowing rivers) in China, (2) an explicit separation of reservoirs from natural lakes among these large water bodies, and (3) the dam points associated with the large reservoirs, with a distinction between river dams and reservoir dams. This is yet another useful data tool in the proliferating global and regional water body datasets.

I explored the produced database, and found the layers well organized and their relationship logically associated. The manuscript is overall clear as well. However, I do have a few major concerns about the concept of dam/reservoir typology, the crosscomparison with other datasets, and some other technical issues I found in China-LDRL. I would like to see a major revision in both text and the dataset that thoroughly addresses the concerns below.

**Reply:** We highly appreciate your insightful comments and suggestions. In this revised version, we first double-checked our lake and reservoir maps in Google Earth one by one, and reclassified those reservoirs that were misclassified as lakes in our previous version, such as Hongze Lake in Jiangsu Province. Second, we renamed the dams as "on-stream" and "off-stream" dams following your suggestion, and rechecked each dam/reservoir and determined its classification. Third, we downloaded the newest version of GeoDAR dataset, which had much better accuracy than the previous version, and re-performed the comparison between newest GeoDAR and our China-LDRL dataset. Finally, we improved the main figures and main text, including the introduction of dam classes (Line 243-249), the comparison between our study and GeoDAR and the discussion of the causes of some of the coordinate offsets in these datasets (Line 330-395).

Dam/reservoir typology

1) The automated extraction of surface water from Landsat images follows a standard mapping pipeline and is technically sound. The separations between natural lakes and reservoirs, and then between the different types of dams, were performed by visually interpreting high-resolution Google Earth images. I don't worry too much on the way the authors identified reservoirs from natural lakes because dams and embankments are often clearly discernible from high-resolution images. My main concern is the classification of the two dam types. First, I am not in favor of the terms "river dams' and 'reservoir dams'. They are a little confusing because many dams on rivers also form reservoirs. At the first glance, I thought a 'river dam' is something like a barrage which has no evident water impoundment, whereas a 'reservoir dam' is the one that impounds reservoirs (either on rivers or not). But this is clearly not what the authors meant after I read the text, especially after I saw the schematic flowchart in Fig. 4. If I understand the authors' intention correctly, I believe a better (and more intuitive) terminology can simply be "on-stream" and "off-stream" dams/reservoirs, with the former constructed on a river/stream (regardless of impoundment) and the latter formed by partial or

complete embankment around an offstream lake (either manmade or originally natural).

**Reply:** Thank you so much for your suggestion. In this revised version, we classified the dams into two types, one is the dams constructed in a river/steam, and another one is those formed by embankment (**Fig. S1**), and we renamed them as "on-stream" and "off-stream" dams following your suggestion. We also introduced the definition of these two types of dams in detail in the main text: "*In an effort to distinguish riverine or off-stream reservoirs from lakes, we uploaded the large SWB vector layers into Google Earth Pro, and checked whether a dam existed around each polygon through the historical satellite images in 2019 within China by visual image interpretation approach. If a dam did not exist, we classified the polygon as river or lake; if a dam does exist, we classified the polygon as on-stream reservoir (constructed on a river/stream regardless of impoundment) or off-stream reservoir (formed by partial or complete embankment around an off-stream lake) (**Fig. S1**). Simultaneously, the corresponding dam would be classified as on-stream dam or off-stream dam. Finally, the SWB polygons were classified into lakes, reservoirs, and rivers, and the dams/reservoirs were classified into on-stream and off-stream dams/reservoirs (**Fig. 3**)*" (Line 240-249).

[Figure]

**Fig. S1.** Types of dams/reservoirs in this study. (a-b) On-stream dam/reservoir constructed on a river/stream regardless of impoundment; (c-d) Off-stream dam/reservoir formed by partial or complete embankment around an off-stream lake. The high-resolution images in this figure were from the Google Earth Pro.

2) Assuming I understand the authors correctly, I found many of the 'reservoir dams' the authors labeled are actually on rivers. Some easy examples are: Zhelin Reservoir on Xiushui River (29.257N, 115.487E), Miyun Reservoir on Chaohe River and Baihe River (40.494N, 116.851E), and Nanwan Reservoir on Shihe River with multiple inflow rivers (32.122N, 114.001E). So I am confused why they were classified as

'reservoir dams'. Was it because some of the rivers are small tributaries that are hard to be seen from Google Earth images? If so, then I encourage that the authors take a deeper stab at the classification, and if necessary, redo some of the classification to ensure a more reliable quality. The revised text should also include a clearer description of the definitions, more detailed rationales and criteria for performing the classification, and relevant limitations of the visual interpretation method.

**Reply:** Yes, we did misclassify the types of some dams/reservoirs due to the classification systems in the previous version. In this revised version, we renamed the dams/reservoirs as "on-stream" and "off-stream" dams/reservoirs following your suggestion, and we have rechecked all these dams/reservoirs in Google Earth to ensure a more reliable quality. In addition, we also introduced the types of dams/reservoirs in the main text (Line 240-249). Furthermore, visual interpretation method also might bring some bout some uncertainties to the classification of dam/reservoir due to the limitation of knowledge and experience of interpreters, and we updated them in the Discussion section: "*In addition, visual interpretation method for identifying dams and reservoirs in this study could also bring about some uncertainties to the classification of dams/reservoirs due to the limitations of knowledge and experience of interpreters, such as the misclassification of some reservoirs regulated by dams/gates as lakes (e.g. Hongze Lake in Jiangsu Province) and the misclassification between on-stream and off-stream dams/reservoirs.*" (Line 422-427).

Comparison with other datasets

3) I overall enjoy reading the comparison section. And I concur that China-LDRL improved the spatial documentation of cascade dams in the South and the Southwest and the reservoirs in the Northeast. Despite the merits, I would like to point out some caveats when attributing coordinate "errors" in the other datasets. Datasets were often produced using different methods and for different purposes. For example in GOODD V1.0, the original digitized dam points were purposefully snapped to the 30-arc-second HydroSHEDS river networks, which led to the offset from the actual dam locations. But on the other hand, GOODD v1.0 is directly compatible with HydroSHEDS and is therefore more convenient for modeling purposes. In GeoDAR v1.1, dam points in China were georeferenced using the Google Maps geocoding API, which led to two "issues". First, the labels for many Chinese dams/reservoirs on Googles Maps are for reservoirs rather than dams (although the names are usually the same). As a result, many "dam" points georeferenced using Google Maps API ended up falling on the reservoir surface instead of on the dams. Second, as the authors should know, Google Maps in China have substantial misalignment (500 m to 1 km or so) between the satellite images and the map labels, because of China's GPS shift problem (which was intentional). This means the geographic coordinates returned from the Google Maps geocoding API will also carry the same offsets, even though the geocoding procedure is correct. This said, the authors may want to fully acknowledge the causes of some of the coordinate "errors" in other datasets, which will warrant a more objective and useful comparison with

China-LDRL.

**Reply:** Thank you for letting us clear the reasons for the misalignment of these datasets. We agree with you that different methods and different purposes could result in the potential offsets, especially, the Google Map in China has substantial misalignment between the satellite images and the geographic data as google maps uses GCJ-02 coordinate system for the street map and labels but uses WGS-84 coordinates for satellite imagery, causing the so-called China GPS shift problem. In our study, we used WGS-84 coordinates for our polygons of surface water body, and identified the large dams within the satellite imagery in Google Earth Pro. The same coordinates in our study could reduce the misalignments between the imagery and dam locations, and improve the geographic accuracy of dams. We also added the reasons for the misalignment of these datasets in the Discussion section of the revised main text (Line 384-395).

4) In addition, the GeoDAR dataset the authors used seems to be an older version. The newest and fully peer-reviewed version of GeoDAR is available at https://doi.org/10.5281/zenodo.6163413 (accepted paper in press). The authors of GeDAR have manually reduced the geographic offsets of many dams in China, so I recommend the authors re-performing the comparison with GeoDAR using its newest version.

**Reply:** We downloaded the newest version of GeoDAR, and checked the dam locations. We found that this newest version had much better accuracy than the previous version although some dams also have offsets due to the GPS shift in China (**Fig. S2**). In addition, we further overlapped the newest GeoDAR layer with our yearlong SWB map, and we found $58.49 \pm 6.07\%$ of its dams were intersected with the yearlong SWB layer, and $82.33 \pm 3.98\%$ and $90.22 \pm 3.18\%$ intersected when the tolerance was 100-m and 500-m (**Fig. S10**). In the revised main text, we reperformed the comparison between our results and newest GeoDAR and discussed the potential reasons for the differences (Line 368-395).

[Figure]

**Fig. S2.** Georeferenced coordinate offsets of dams from the GeoDAR within high-resolution images.

[Figure]

**Fig. S10.** Numbers of dams intersected with surface water body (SWB) map in China for 2019 by province. (a) Percentage of numbers of dams intersected with SWB map; (b) Percentage of numbers of dams intersected with SWB map when applying a 100-m tolerance.

Other technical issues

5) I would like to point out a few other technical issues I found in China-LDRL. Some of the natural lakes the authors classified are actually reservoirs or regulated (dammed/gated) lakes. Examples are lake ID 18395, which is part of the Danjingkou Reservoir, and lake IDs 1904, 2176, 1428, and 1483, which are arguably part of the Three Gorges Reservoir stretching to the lateral tributaries (some of the polygons were cut off by bridges), and lake ID 76265, which duplicates (conflicts) with reservoir ID 1261.The authors mentioned both GRanD and GeoDAR misidentified Hongze Lake as a reservoir. After a careful examination, I believe Hongze Lake should be a regulated lake (controlled by Sanhezha Gate), thus compliant with the reservoir category (by the way, lake ID 90753 seems to be an editing glitch). In general, I encourage the authors to perform another round of quality control on the classified natural lakes and reservoirs to ensure the accuracy as much as they can.

**Reply:** You raised very good problems in our dataset. We did classify some reservoirs as natural lakes because of their names, such as the Hongze Lake in Jiangsu Province and Dongchang Lake and Dongping Lake in Shandong Province. In this revised version, we double-checked these lake polygons one by one, and tried to make sure these lakes were classified correctly. In addition, as we used the yearlong surface water body to generate the lake/reservoir maps and some polygons were cut off into several polygons by bridges or provincial boundaries, in this revised version, we tried to recheck each polygon and combined potential polygons as one polygon with one same ID when they were regarded as one lake or reservoir. After the checking of each polygon, we recalculated the areas of lake and reservoir, and reassigned an ID for each polygon to make sure there is no conflicts.

6) I agree with Lines 105 to 111 that water body classes have not been adequately considered in water body dynamics studies. This echoes the PNAS letter from Song et al. (https://doi.org/10.1073/pnas.2005584117), which demonstrates the importance of water body types in sorting out the recent surface water dynamics in China.

**Reply:** Song et al. (2020) did a nice work of mapping lakes in China and demonstrating the importance of lakes and reservoirs. According to Song's study, we introduced the importance and necessary to distinguish lake and reservoirs from surface water body in the Introduction section (Line 37-54), and discussed the potential of our study for exploring the dynamics of different water body types and the conversions between lakes and reservoirs (Line 434-444).

---

## Author Response (AR2)

**Comments from the Editor:**
Thank you for using ESSD!

Small issue, can resolve at proof stage: The disputed Spratly Islands inset shows up occasionally: Fig 2f, Fig 4c, Fig 7d, Fig 8f, Fig 9d. Because this study includes no SWB from those regions, authors should omit those insets? If retained, those insets will trigger a standard Copernicus disclaimer attached to this manuscript, declaring the geography and political governance of those regions as under discussion.

**Response:**
Thank you very much for your comments on our manuscript. With regards to the Spratly Islands insets in the figures, we would like to keep them as they stand. We are okay to trigger a standard Copernicus disclaimer attached to this manuscript, if needed.